# Joint association of sedentary behavior and physical activity domains with depression in Korean adults: Cross-sectional study combining four biennial surveys (2016–2022)

**Sungjin Park** [1], **June-Hee Lee** [2]*

**1** Department of Occupational and Environmental Medicine, Gwanghwamun Center, Korea Medical Institute, Seoul, Republic of Korea, **2** Department of Occupational and Environmental Medicine, Soonchunhyang University Hospital, Seoul, Republic of Korea

* junelee@sch.ac.kr

**Data Availability Statement:** The KNHANES is openly available to use publicly on the following address. https://knhanes.kdca.go.kr/knhanes/sub03/sub03_02_05.do.

## Abstract

Although the increased prevalence of sedentary behavior and insufficient physical activity constitutes a global public health concern, there is limited research on their effects on mental health. We investigated the combined association of sedentary behavior (daily sitting or reclining ≥10 h/day) and physical activity domains (evaluated using the Global Physical Activity Questionnaire, including occupational physical activity, leisure-time physical activity, and transportation-related physical activity) with depression (Patient Health Questionnaire-9, cutoff score: 10). This cross-sectional study utilized biennial data of 21,416 adults (age >20 years) from the Korea National Health and Nutrition Examination Survey waves 7–9 (2016–2022). Joint associations were explored by combining sedentary behavior and each physical activity domain into four levels. Sedentary behavior and occupational physical activity increased the risk of depression, leisure-time physical activity decreased the risk only in men, and transportation-related physical activity showed no significant association. Logistic regression each physical activity domain revealed, for men and women, a significantly higher risk of depression in the sedentary behavior (+)/occupational physical activity (+) group than in the sedentary behavior (–)/occupational physical activity (–) group (odds ratio: 3.05 and 2.66, respectively). The sedentary-behavior (+)/leisure-time physical-activity (–) group showed a significantly higher risk of depression than the sedentary behavior (–)/leisure-time physical activity (+) group (odds ratio: 2.50 and 2.14), and sedentary behavior (+)/transportation-related physical activity (–) group also showed a significantly higher risk of depression compared to the sedentary behavior (–)/transportation-related physical activity (+) group (odds ratio: 1.83 and 1.61). With concurrent exposure to sedentary behavior, the occupational physical activity and lack of leisure time and transportation-related physical activity synergistically increased the risk of depression. Encouraging leisure-time physical activity, minimizing rigorous occupational physical activity, and reducing sedentary behavior may reduce depressive symptoms, and research into specific domains of sedentary behavior and the quantity and quality of transportation-related physical activity is needed.

**Funding:** This study was supported by the Soonchunhyang University Research Fund. The funders had no role in study design, data collection and analysis, decision to publish, or preparation of the manuscript.

**Competing interests:** The authors have declared that no competing interests exist.

## Introduction

Individuals are extensively exposed to sedentary behavior (SB), which is defined as any waking activity with an energy expenditure of less than 1.5 metabolic equivalent tasks, such as watching television, playing video games, using computers, sitting at school or work, and commuting while seated [1]. Recent trends indicate an increasing prevalence of SB, and a systematic review of 26 studies revealed an increase in sedentary lifestyles worldwide [2]. The prevalence of remaining seated for 8 or more hours increased significantly from 46.7%, in 2014, to 56.2%, in 2017, and 63%, in 2021 [3]. Increased SB represents a serious public health issue, [4] as sedentary lifestyles are associated with various adverse health outcomes, including increased all-cause and cardiovascular disease (CVD) mortality; [5] metabolic diseases, [6, 7] such as diabetes mellitus (DM), hypertension (HTN), and dyslipidemia; inflammatory markers; [8] and cognitive impairment [1].

Health research on SB should consider physical activity (PA), [9] as SB and PA are closely interrelated and are not mutually exclusive. For example, PA mitigates SB and prevents its persistence [4]. Recent studies have extensively explored the impacts of PA, particularly on CVD [10–13]. Although leisure-time PA (LTPA) or total PA has traditionally recognized benefits, the negative health effects of occupational PA (OPA) or household PA (HPA) indicate the phenomenon of the PA paradox [14].

Furthermore, in addition to physical health, the relationship between mental health and the SB and PA domains has been examined. Increased SB has a dose–response relationship with frequent depressive symptoms, [15] whereas replacing sedentary time at work with vigorous PA decreases the risk of depression [16]. Moreover, LTPA is associated with decreased incident rates of depression [17]. However, the association of PA domains, other than LTPA, with mental health remains unclear. In several studies, OPA was associated with improved mental health [18] whereas, conversely, two reviews reported associations of high OPA levels with increased depression and anxiety, [19, 20] and a Japanese cohort study reported lower rates of depression in more active occupations, compared with those in sedentary occupations [21]. In addition, there is limited research on transportation-related PA (TRPA) or active commuting.

The inconsistencies in the association of various domains of PA, except LTPA, with depression underscore the need for more evidence. Moreover, studies on the health effects of PA domains in different SB levels, particularly on their association with depression, are lacking [19]. Therefore, in this study, we aimed to investigate the relationship between the risk of depression and combinations of three PA domains (OPA, LTPA, and TRPA) with SB by using national data that represent the health status and physical activity levels of the Korean general population.

## Methods

### Data description and participants

This cross-sectional study, we used data from the 7th through 9th waves (2016–2022) of the Korea National Health and Nutrition Examination Survey (KNHANES), which is implemented by the Korea Centers for Disease Control and Prevention to monitor the health and nutritional status of South Koreans and collect statistical data. Demographic and socioeconomic characteristics, such as age, sex, marital status, education level, monthly income, occupation, health behaviors, and personal or family medical history, are collected through interview-based surveys. KNHANES obtained approval from the Institutional Review Board of the Korea Centers for Disease Control and Prevention, and written informed consent was collected from all the participants. Ethical approval was not required for this analysis because

this research entailed a secondary analysis of publicly accessible Korean national data without personal identifiers.

The Patient Health Questionnaire-9 (PHQ-9), a standardized depression rating scale, has been administered every 2 years since 2014 [22]. Therefore, after excluding the outdated data from 2014, data from 2016, 2018, 2020, and 2022 were combined to enhance the statistical power. Among the 29,766 individuals surveyed over the 4 years, 21,873 had available PHQ-9 data (with 7,893 missing). After excluding 242 individuals with missing data on SB and 215 teenagers (age <20 years), the final analysis included 21,416 adults.

## Measurements

SB was assessed by asking the participants about their daily sitting or reclining hours, and was defined as ≥10 hours of sitting or reclining per day, as determined by previous epidemiologic studies [23–26]. PA was evaluated using the Global Physical Activity Questionnaire (GPAQ), which is a component of the KNHANES. Originally developed by the World Health Organization (WHO), this self-administered instrument assesses OPA, LTPA, and TRPA [27]. The reliability and validity of the Korean version of the GPAQ have been verified [28]. In the GPAQ, "vigorous-intensity activity" refers to PA that involves rigorous physical exertion and a significant increase in breathing or heart rate, whereas "moderate-intensity activity" refers to PA that involves moderate physical exertion and a modest increase in breathing or heart rate. The WHO guidelines recommend at least 150 or 75 min of moderate-intensity or vigorous-intensity activity, respectively, every week [29]. Individuals who met the quantitative criteria for moderate-to-vigorous PA (MVPA) at work or during leisure-time activities were classified as having OPA or LTPA, respectively [29]. TRPA was defined as involving at least 150 min per week of physical activities, such as walking or biking when traveling between locations, excluding work-related travel.

The combination of each PA with SB generated four levels of joint association. For activities expected to reduce the risk, such as LTPA and TPA, the groups were categorized as those with: 1) low SB/high PA (reference group), 2) high SB/high PA, 3) low SB/low PA, and 4) high SB/low PA. Conversely, for activities that potentially increased the risk of depression, such as OPA, the groups were classified as: 1) low SB/low OPA (reference group), 2) high SB/low OPA, 3) low SB/high OPA, and 4) high SB/high OPA.

The KNHANES VII–IX employed the PHQ-9 to assess depressive symptoms. This self-report tool is a reliable screening scale for identifying depressive symptoms. PHQ-9 rates nine items on a scale of 0 to 3 (not at all = 0, several days = 1, more than half of the days = 2, and nearly every day = 3), which indicate symptom frequency over the past 2 weeks, as aligned with the criteria of *Diagnostic and Statistical Manual of Mental Disorders, 4th edition* (DSM-IV) [29]. The cumulative score (0–27) denotes symptom severity, with higher scores indicating greater severity [30]. A cutoff score of ≥10 indicates the presence of depression, which potentially requires treatment, including medication [31, 32]. This cutoff shows adequate sensitivity (88%) and specificity (88%) for identifying depression [33].

This study encompassed various socioeconomic and health-related factors, including sex (men or women), age group (20s, 30s, 40s, 50s, 60 or older), marital status (married, divorced, single, widowed), obesity (BMI ≥25 kg/m$^2$), smoking status (current smoker, former/non-smoker), heavy alcohol consumption, income level, educational level (middle school or lower, high school, college or higher), occupation type (white-collar, pink-collar in sales or service, blue-collar), and presence of chronic disease (HTN, DM, dyslipidemia, and CVD), as diagnosed by a physician. Income level was classified into quartiles based on annual income, ranging from low to high. Heavy alcohol consumption was defined as the intake of ≥7 and ≥5 drinks, for men and women, respectively, on ≥2 occasions per week in the past year.

## Statistical analysis

The association between depression risk and participants' demographic and health characteristics was examined using the chi-square test. Multiple logistic regression analyses were performed to determine the odds ratios (ORs) and 95% confidence intervals (CIs) of the SB and PA domains in relation to the risk of depression. These analyses included the variables that were significantly associated with depression, including sex, age, marital status, obesity, smoking status, heavy drinking, income level, education level, and chronic diseases. Furthermore, to explore the joint associations of each PA–SB domain combination with the risk of depression, the ORs (95% CIs) were computed across the four groups. All the analyses were stratified by sex. All statistical analyses were performed using IBM SPSS Statistics for Windows ver. 27.0 (IBM Corp., Armonk, NY, USA). The statistical significance was set at $p < 0.05$.

## Results

Table 1 shows the results of the chi-square test for the association of the prevalence of depression with socioeconomic characteristics, SB, and PA domains for men and women. The increased prevalence of depression was significantly associated with being women; being divorced, single, widowed, obese, current smoker, heavy drinker; having lower education or income levels; and the presence of chronic disease. The prevalence of depression exhibited a U-shaped relationship with age and was the highest among pink-collar workers, followed by blue- and white-collar workers. Compared to their counterparts, individuals who were SB(+), OPA(+), and LTPA(–) had a significantly higher prevalence of depression (6.52% vs. 4.27%, 8.24% vs. 5%, and 5.55% vs. 3.2%, respectively). Although the prevalence was higher in TRPA (–) individuals (5.31% vs. 4.83%), this was not statistically significant. The overall results for all participants were similar when stratified by sex, except for the fact that, among men, no significant association with age was detected, and the prevalence of depression was higher among blue-collar workers than those in pink-collar workers.

Table 2 presents the adjusted ORs [95% CI] for the association of the combination of SB and different types of PA with the risk of depression. In men, after adjusting for all the variables that were significantly associated with the prevalence of depression, the risk of depression significantly increased with SB (1.83 [1.46–2.30]) and OPA (1.82 [1.26–2.63]) and decreased with LTPA (0.60 [0.42–0.84]). Similarly, in women, the risk of depression significantly increased with SB (1.51 [1.29–1.77]) and OPA (2.29 [1.70–3.08]) and non-significantly decreased with LTPA (0.81 [0.62–1.06]) and TRPA (0.88 [0.74–1.05]).

Table 3 outlines the combined association of each PA domain–SB with the risk of depression, and Figs 1 and 2 present the ORs [95% CIs] for men and women, respectively. Compared with the reference group with SB (–)/OPA (–), the SB(+)/OPA(+) group had the highest risk of depression (3.05 [1.62–5.76] and 2.66 [1.53–4.63], respectively). Compared with the reference group with SB(–)/LTPA(+), in men, the SB (+)/LTPA(–) group had the highest risk of (2.50 [1.57–3.99]), whereas, in women, the SB(+)/LTPA(+) (2.15 [1.30–3.56]) and SB (+)/LTPA(–) (2.14 [1.45–3.15]) groups had a significantly increased risk of depression. Finally, in women, the SB(+)/TRPA(–) group (1.61 [1.28–2.03]) and, in men, the SB(+)/TRPA(+) (2.13 [1.39–3.28]) and SB(+)/TRPA(–) groups (1.83 [1.28–2.61]) had significantly increased risk for depression.

## Discussion

In this study, we categorized PA into OPA, TPA, and LPA and examined the association of SB and each PA type with the prevalence of depression and ascertained their joint association with the risk of depression. Unlike previous research that focused on the relationship between

**Table 1. Association of demographics with depression in Korean adults (2016–2022).**

| Characteristics | Prevalence of Depression [N (%)] | | |
| --- | --- | --- | --- |
| | Total (N = 20673) | Men (N = 9397) | Women (N = 12019) |
| **Sex** | 1109 (5.18) | 344 (3.66) | 765 (6.36)** |
| **Age, years** | ** | NS | ** |
| 20–29 | 179 (7.03) | 53 (4.48) | 126 (9.25) |
| 30–39 | 182 (5.61) | 62 (4.33) | 120 (6.63) |
| 40–49 | 161 (4.16) | 63 (3.84) | 98 (4.4) |
| 50–59 | 154 (3.83) | 46 (2.71) | 108 (4.65) |
| ≥60 | 433 (5.59) | 120 (3.49) | 313 (7.28) |
| **Marital status** | ** | ** | ** |
| Married | 834 (4.73) | 226 (3.06) | 608 (5.93) |
| Other status | 275 (7.28) | 118 (5.87) | 157 (8.87) |
| **Body mass index, kg/m²** | ** | ** | ** |
| <25 | 1002 (4.98) | 311 (3.54) | 691 (6.09) |
| ≥25 | 107 (8.26) | 33 (5.35) | 74 (10.91) |
| **Smoking status** | ** | ** | ** |
| Nonsmoker or ex-smoker | 816 (4.61) | 180 (2.84) | 636 (5.6) |
| Current smoker | 290 (7.88) | 164 (5.38) | 126 (19.97) |
| **Heavy drinking** | ** | ** | ** |
| No | 937 (4.96) | 248 (3.29) | 689 (6.06) |
| Yes | 169 (6.83) | 96 (5.21) | 73 (11.55) |
| **Education level** | ** | ** | ** |
| Middle school or lower | 6044 (7.51) | 107 (5.03) | 347 (8.86) |
| High school | 352 (4.99) | 120 (3.66) | 232 (6.15) |
| College or higher | 302 (3.63) | 116 (2.91) | 186 (4.3) |
| **Income level** | ** | ** | ** |
| Low | 452 (8.7) | 166 (7.24) | 286 (9.86) |
| Medium-low | 296 (5.51) | 77 (3.29) | 219 (7.23) |
| Medium-high | 205 (3.82) | 51 (2.16) | 154 (5.12) |
| High | 155 (2.85) | 50 (2.1) | 105 (3.44) |
| **Occupation** | ** | ** | ** |
| White collar | 300 (3.87) | 95 (2.85) | 205 (4.65) |
| Pink collar | 309 (6.99) | 56 (4.08) | 253 (8.3) |
| Blue collar | 364 (5.58) | 151 (4.31) | 213 (7.05) |
| **Chronic disease[a]** | ** | ** | ** |
| No | 638 (4.51) | 201 (3.32) | 437 (5.4) |
| Yes | 471 (6.48) | 143 (4.28) | 328 (8.35) |
| **SB[b]** | ** | ** | ** |
| No | 544 (4.27) | 162 (2.93) | 382 (5.29) |
| Yes | 565 (6.52) | 182 (4.7) | 383 (7.98) |
| **OPA[c]** | ** | ** | ** |
| No | 1011 (5) | 306 (3.51) | 705 (6.13) |
| Yes | 98 (8.24) | 38 (5.6) | 60 (11.76) |
| **LTPA[d]** | ** | ** | ** |
| No | 1001 (5.55) | 303 (4.04) | 698 (6.62) |
| Yes | 108 (3.2) | 41 (2.17) | 67 (4.52) |
| **TRPA[e]** | NS | NS | NS |
| No | 828 (5.31) | 252 (3.65) | 576 (6.62) |

(*Continued*)

**Table 1.** (Continued)

| Characteristics | Prevalence of Depression [N (%)] | | |
|---|---|---|---|
| | Total (N = 20673) | Men (N = 9397) | Women (N = 12019) |
| Yes | 281 (4.83) | 92 (3.68) | 189 (5.69) |

SB = sedentary behavior, PA = physical activity, OPA = occupational physical activity, LTPA = leisure-time physical activity, TRPA = transportation-related physical activity.

*$p<0.05$ and

**$p<0.01$, from chi-square tests, between the prevalence of depression and characteristics.

[a]Having any chronic diseases, including hypertension, diabetes, dyslipidemia, and cardiovascular disease, diagnosed by a physician.

[b]Sitting time ≥10 h/day.

[c]Undertaking ≥150 or ≥75 minutes of moderate or vigorous PA, respectively, per week at work.

[d]Undertaking ≥150 or ≥75 minutes of moderate or vigorous PA, respectively, per week during leisure-time.

[e]Undertaking ≥150 minutes of PA per week during transportation.

either PA or SB alone and depression, this study uniquely explores the joint association of specific PA types and SB with depression. The findings suggest that, depending on the PA domains, various SB–PA combinations can have different impacts on mental health.

The main findings of this study are twofold. First, in men and women, SB and OPA were significantly associated with an increased risk of depression. LTPA was significantly associated with a reduced risk of depression only in men, whereas LTPA and TRPA were associated with a reduced risk of depression in women. Second, the analysis of each PA domain–SB combination revealed that, in men and women, compared with the SB(−)/OPA(−) reference group, the SB(+)/OPA(+) group had the highest risk of depression. Despite sex-specific differences, compared with the SB(−)/LTPA(+) reference group, the SB(+)/LTPA(−) group had a significantly higher risk of depression, and the same trend was observed for the SB(−)/TRPA(+) group. This finding is meaningful as it indicates that LTPA (in women) and TRPA (in men and women), which were not individually associated with the risk of depression, were significantly associated with this risk when simultaneously exposed to SB(+); this suggests a synergistic effect of these factors on the risk of depression. Furthermore, in women, the SB(+)/LTPA(+) group showed the highest increase in the risk of depression (OR = 2.15), which indicates SB's stronger association with the risk of depression as compared with that of LTPA. Likewise, in men, the SB(+)/TRPA(+) group had the highest increase in the risk of depression (OR = 2.13), indicating SB's dominant association, compared to that of TRPA, with the risk of depression.

**Table 2. Individual association of SB and PA domains with depression in Korean adults (2016–2022).**

| Variables | Adjusted[a] odds ratios and 95% confidence intervals | |
|---|---|---|
| | Men | Women |
| SB (Yes vs. No) | 1.83 (1.46–2.30) | 1.51 (1.29–1.77) |
| OPA (Yes vs. No) | 1.82 (1.26–2.63) | 2.29 (1.70–3.08) |
| LTPA (Yes vs. No) | 0.60 (0.42–0.84) | 0.81 (0.62–1.06) |
| TRPA (Yes vs. No) | 1.05 (0.81–1.35) | 0.88 (0.74–1.05) |

OR = odds ratio, CI = confidence interval, SB = sedentary behavior, PA = physical activity, OPA = occupational physical activity, LTPA = leisure-time physical activity, TRPA = transportation-related physical activity.

[a]Adjusted covariates included age, marital status, obesity, smoking status, heavy drinking, income level, educational level, occupational type, having chronic diseases, and SB, OPA, LTPA, and TRPA.

**Table 3. Joint association of each PA domain–SB with depression in Korean adults (2016–2022).**

| Combination of SB and PA | Adjusted[a] odds ratios and 95% confidence intervals | |
|---|---|---|
| | Men | Women |
| **SB (–)/OPA (–)** | 1.00 (reference) | 1.00 (reference) |
| **SB (+)/OPA (–)** | 1.83 (1.44–2.32) | 1.6 (1.36–1.88) |
| **SB (–)/OPA (+)** | 1.76 (1.13–2.73) | 2.59 (1.83–3.67) |
| **SB (+)/OPA (+)** | 3.05 (1.62–5.76) | 2.66 (1.53–4.63) |
| **SB (–)/LTPA (+)** | 1.00 (reference) | 1.00 (reference) |
| **SB (+)/LTPA (+)** | 1.36 (0.73–2.54) | 2.15 (1.30–3.56) |
| **SB (–)/LTPA (–)** | 1.40 (0.87–2.25) | 1.48 (1.01–2.18) |
| **SB (+)/LTPA (–)** | 2.50 (1.57–3.99) | 2.14 (1.45–3.15) |
| **SB (–)/TRPA (+)** | 1.00 (reference) | 1.00 (reference) |
| **SB (+)/TRPA (+)** | 2.13 (1.39–3.28) | 1.26 (0.92–1.71) |
| **SB (–)/TRPA (–)** | 1.12 (0.78–1.61) | 1.02 (0.81–1.29) |
| **SB (+)/TRPA (–)** | 1.83 (1.28–2.61) | 1.61 (1.28–2.03) |

OR = odds ratio, CI = confidence interval, SB = sedentary behavior, PA = physical activity, OPA = occupational physical activity, LTPA = leisure-time physical activity, TRPA = transportation-related physical activity.
[a]Adjusted covariates included age, marital status, obesity, smoking status, heavy drinking, income level, educational level, occupational type, having chronic diseases, and SB, OPA, LTPA, and TRPA.

LTPA and SB were significantly associated with depression in previous systematic reviews and meta-analyses of prospective studies [34–36]. SB has been consistently associated with increased prevalence and incidence rates of depression across domains, whereas LTPA protects against the risk of depression. Plausible explanations for this primarily revolve around traditional LTPA, namely exercise, which offers psychological benefits, such as distraction from negative thoughts, and improved physical self-perceptions and self-efficacy. Additionally, social benefits, such as social support and participation, contribute to social reinforcement, whereas the physiological benefits include increased endorphin levels and neurotransmitter availability [21, 35, 37].

Prolonged SB may elevate the risk of depression by displacing time spent on PA, which can alleviate depressive symptoms [36]. Particularly, excessive television watching and use of personal computers and smartphones can increase the risk of depression by disrupting direct interpersonal communication and social interaction, in what is known as the social displacement hypothesis [38]. Korea is a global leader in smartphone and internet usage, with widespread access in workplaces and daily life [3]. Reducing the use of smart or mobile devices can reduce SB, and thereby influence PA and its contribution to depression, especially in Korea.

The association of OPA with an increased risk of depression in this study aligns with the results of previous research. A recent meta-analysis indicated a positive association between OPA and ill mental health, [10] and a study using KNHANES data reported that OPA was not as beneficial as LTPA and significantly increased depressive symptoms [39]. OPA differs fundamentally from LTPA regarding motivation, locus of control, and enjoyment. Regarding LTPA, motivation refers to weight control, fitness, and mood improvement; locus of control refers to autonomy in choosing the type, intensity, location, and timing of PA; and enjoyment is a key factor in the positive psychological outcomes of exercise. Although these aspects play a crucial role in the antidepressant effects of LTPA, they are not shared by OPA. Instead, intrinsic motivation for OPA is usually driven by financial gain, and OPA is associated with lower levels of control and autonomy [40]. Finally, despite its potential contribution to overall PA

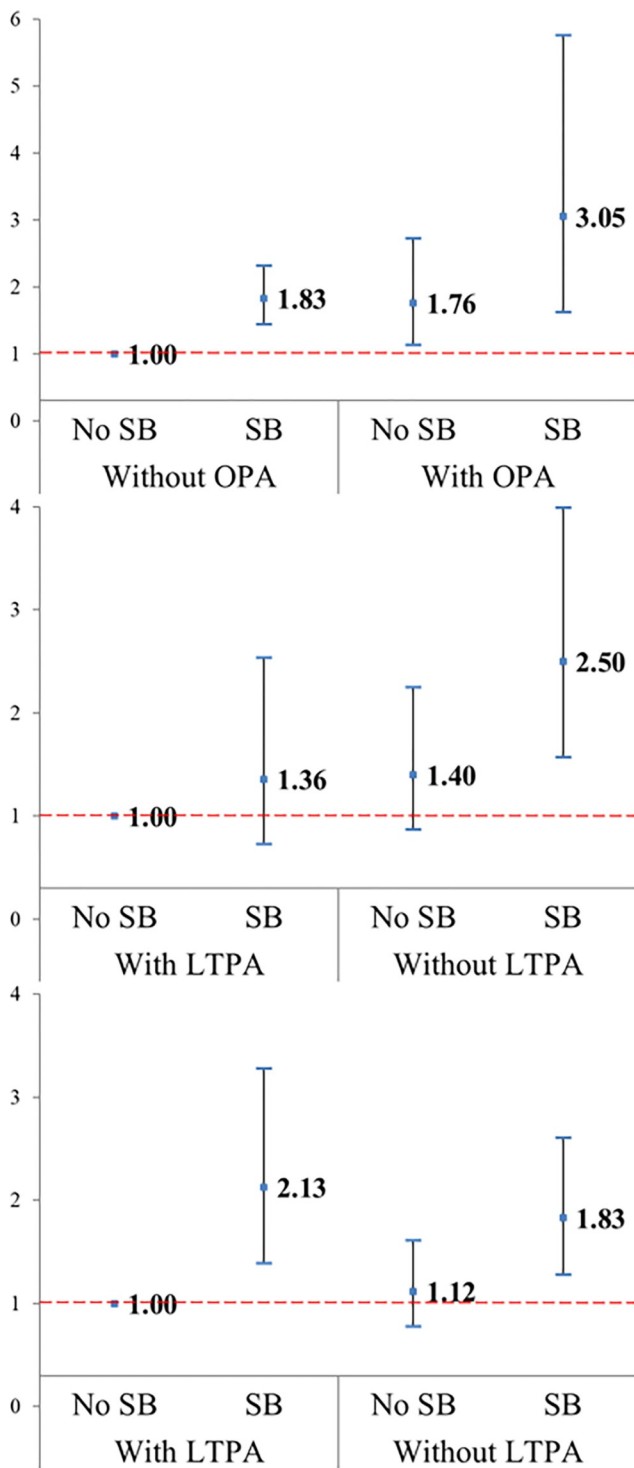

**Fig 1. Fully adjusted odds ratios and 95% confidence intervals for depression risks in association with sedentary behavior (SB) and physical activity (PA) domains, including occupational PA (OPA), leisure-time PA (LTPA), or transportation-related PA (TRPA) in men.**

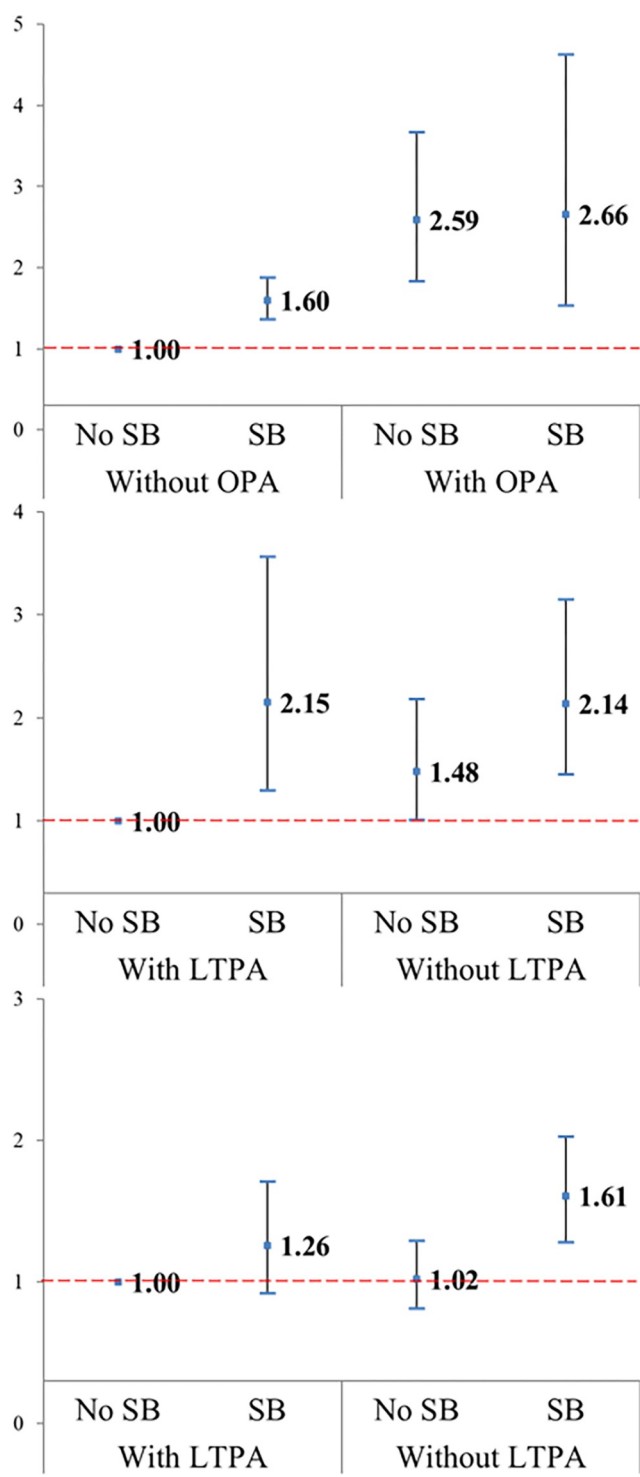

**Fig 2. Fully adjusted odds ratios and 95% confidence intervals for depression risks in association with sedentary behavior (SB) and physical activity (PA) domains, including occupational PA (OPA), leisure-time PA (LTPA), or transportation-related PA (TRPA) in women.**

levels, high OPA was associated with minority ethnic groups and lower socioeconomic status, [41] which are associated with an increased risk of depression [42].

The association of TRPA with depression was only marginally significant in women and, when combined with SB, weakened further. Previous research on the relationship between TRPA and depressive symptoms has predominantly yielded null findings [21, 43]. Moreover, a more recent systematic review reported inconclusive results, with active commuting having beneficial effects on depressive symptoms in only 2 of 7 studies [44]. Theoretically, TRPA, such as active commuting, can reduce depression risk by contributing to an increase in daily PA levels and serve as a valuable source of PA for several workers with limited time for exercise outside of commuting. Research is warranted to further explore this relationship, and future studies should conduct more detailed analyses of the types, frequency, and volume of TRPA.

The focus of this study was to analyze the previously unexplored association of the combination of SB and each PA domain with depression, and this enhances the significance of this study. Individuals can be both physically active and highly sedentary, as these are not mutually exclusive [45]. Particularly, the increased risk of depression was prominent when individuals were exposed to both OPA and SB, and this suggests that OPA may not necessarily have the same positive effects as exercise [46]. For example, blue-collar workers involved in OPA during work hours may also exhibit SB at home, whereas pink- or white-collar workers, who are predominantly engaged in SB at work, may perform OPA as part of their job duties, and potentially increases the risk of depression owing to the simultaneous exposure to OPA and SB. The prevalence of OPA(+) individuals varied across the occupational groups, with the highest proportion observed among blue-collar workers (5.38%). However, some level of OPA was observed even in the occupations characterized by prolonged SB (white-collar workers, 5.02%; pink-collar workers, 4.25%). Therefore, as preventive measures against depression, MVPA should be reduced at work or substituted with lighter PA and lower SB.

The increased risk of depression in the LTPA(−)/SB(+) group in this study supports previous findings of the combined association of SB and overall PA with the risk of depression [34, 47]. The lack of domain-specific SB measurement could be a reason for the nonsignificant association of SB at different PA levels and the risk of depression in a study conducted in Japan [37]. Detailed information about situations where SB occurs, such as workplaces or homes, may facilitate a better understanding of the temporal relationships between PA and SB. Particularly in Korea, where the percentage of SB has been increasing annually, future national surveys should include assessments of SB in different domains to facilitate its utilization in research.

## Limitations

This study has some limitations. First, owing to the nature of mental health conditions, this study's reliance on data from cross-sectional surveys may have resulted in a cyclical and reciprocal association between increased SB and decreased PA with regard to the risk of depression. Second, sitting time was assessed with a single question, which raises issues regarding validation. Moreover, the SB domain was not specifically investigated, which limits the application and interpretation of the results. Future research should explore the SB domains in detail. Finally, the use of data from 2016 to 2022 may have introduced variations in the rates of depression and PA across the years. Although the difference was expected, particularly with the onset of the coronavirus disease pandemic in 2020, additional analyses that incorporate the survey year as a covariate did not significantly alter the significance or trends of the results.

## Conclusion

SB and OPA contribute to the risk of depression, whereas LTPA exhibits a protective effect. However, a protective effect of TRPA on depressive symptoms was not observed. Encouraging PA during leisure time, avoiding excessive PA at work, and reducing SB may help reduce depressive symptoms. Future research should clarify the specific domains of SB and the detailed aspects of TRPA in relation to the risk of depression.

## Acknowledgments

The authors would like to thank Korea Medical Institute for their support of the article.

## Author Contributions

**Conceptualization:** Sungjin Park.

**Data curation:** Sungjin Park.

**Formal analysis:** Sungjin Park, June-Hee Lee.

**Funding acquisition:** Sungjin Park, June-Hee Lee.

**Investigation:** Sungjin Park.

**Methodology:** Sungjin Park, June-Hee Lee.

**Project administration:** Sungjin Park.

**Resources:** Sungjin Park.

**Software:** Sungjin Park, June-Hee Lee.

**Supervision:** Sungjin Park, June-Hee Lee.

**Validation:** Sungjin Park, June-Hee Lee.

**Visualization:** Sungjin Park, June-Hee Lee.

**Writing – original draft:** Sungjin Park.

**Writing – review & editing:** Sungjin Park, June-Hee Lee.

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
