## [Decision Letter · Decision Letter 0]

8 Aug 2024

PONE-D-24-21184Joint association of sedentary behaviors and physical activity domains with depression in Korean adults: Cross-sectional study using four biennial surveys (2016–2022)PLOS ONE

Dear Dr. Park,

Thank you for submitting your manuscript to PLOS ONE. After careful consideration, we feel that it has merit but does not fully meet PLOS ONE’s publication criteria as it currently stands. Therefore, we invite you to submit a revised version of the manuscript that addresses the points raised during the review process.

We look forward to receiving your revised manuscript.

Kind regards,

Rashid Menhas, PhD

Academic Editor

PLOS ONE

Journal Requirements:

Reviewers' comments:

Reviewer's Responses to Questions

**Comments to the Author**

1. Is the manuscript technically sound, and do the data support the conclusions?

Reviewer #1: Partly

Reviewer #2: Partly

2. Has the statistical analysis been performed appropriately and rigorously? 

Reviewer #1: Yes

Reviewer #2: Yes

3. Have the authors made all data underlying the findings in their manuscript fully available?

Reviewer #1: Yes

Reviewer #2: Yes

4. Is the manuscript presented in an intelligible fashion and written in standard English?

Reviewer #1: Yes

Reviewer #2: Yes

5. Review Comments to the Author

Reviewer #1: The article topic is very interesting and explores the Joint association of sedentary behaviors and physical activity domains with depression in Korean adults: Cross-sectional study using four biennial surveys (2016–2022). The scholar appears to possess a reasonable level of research temperament and knowledge of her/his area of study and seems comfortable with the subject understanding. However, I deem it appropriate to mention some major observations about different dimensions of the work and, therefore, recommend the following changes to be made before the decision.

Abstract.

• Sometimes written as an afterthought, the abstract is of extreme importance as in many instances this section is what is initially previewed by readers to determine if the remainder of the paper is worth reading. This is the author's opportunity to draw the reader into the study and entice them to read the rest of the article.

• The abstract is a summary of the study and allows the readers to get a glance at what the contents of the article include.

• Writing an abstract is rather challenging as being brief, accurate, and concise are requisite.

• The headings and structure for an abstract are usually provided in the instructions for authors.

Introduction and Review of Literature

• The introduction is one of the more difficult portions of the manuscript to write. Past studies are used to set the stage or provide the reader with information regarding the necessity of the

represented project.

For an introduction to work properly, the reader must feel that the research question is clear, concise, and worthy of study.

• A competent introduction should include at least four key concepts: 1) the significance of the topic, the information gap in the available literature associated with the topic, 3) a literature review in support of the key questions, and 4) subsequently developed purposes/objectives.

Methods

• Initially a brief paragraph should explain the overall procedures and study design. The methods section should clearly describe the specific design of the study and provide a clear

and concise description of the procedures that were performed. The purpose of sufficient detail in the methods section is so that an appropriately trained person would be able to replicate your experiments.

• There should be complete transparency when describing the study

• A clear methods section should contain the following information: 1) the population and equipment used in the study, 2) how the population and equipment were prepared and what was done during the study, 3) the protocol used, 4) the outcomes and how they were measured, 5) the methods used for data analysis.

Result and Discussion

• It should state the impact of your results compared with recent work and relate it to the problem or question you posed in your introduction. Ensure claims are backed up by evidence and explain any complex arguments.

• Writing for the thesis can be a challenging yet satisfying endeavor.

• The ability to examine, relate, and interlink evidence, as well as to provide a peer-reviewed, disseminated product of your research labor can be rewarding.

• This is for interpretation of the key results and to highlight the novelty and significance of the work.

However, the authors are advised to address the following issues before the final submission and highlight in red for double checking.

• There are certain vague and redundant expressions.

• Incorrect use of punctuation marks has been observed. It is compulsory to update the Literature review and discussion section and cite these studies to improve the quality of the literature review and discussion.

In introduction/ literature review section and discussion section have serious flows, authors are advised to follow above comments to update these sections very carefully and cite these papers in these sections to improve the quality of this manuscript. All revisions should be highlighted in red for double checking.

1. https://doi.org/10.2147/PRBM.S441395

2. https://doi.org/10.3389/fpubh.2023.1252157

3. https://doi.org/10.2147/PRBM.S405273

4. https://doi.org/10.3389/fpsyg.2021.667461

5. https://doi.org/10.2147/RMHP.S258660

6. https://doi.org/10.3389/fpsyg.2020.614770

7. https://doi.org/10.3389/fpsyg.2022.933974

8. https://doi.org/10.3389/fpsyg.2022.948061

9. https://doi.org/10.2147/PRBM.S441395

10. https://doi.org/10.2147/PRBM.S369020

• A careful proofreading is required.

Reviewer #2: The purpose of this study was to investigate the combined associations of Sedentary Behavior and Physical Activity domains with depression. This cross-sectional research study included 21,416 adults older than 20 years and

utilized biennial data from the Korea National Health and Nutrition Examination Survey waves 7 to 9 (2016–2022). SB was assessed using daily sitting or reclining hours, defined as ≥10 hours per day. PA was evaluated using the Global Physical Activity Questionnaire (GPAQ), including occupational physical activity (OPA), leisure-time physical activity (LTPA), and transportation-related physical activity (TRPA). In addition, Depressive symptoms were measured using the Patient Health Questionnaire-9, with a cutoff score of 10 indicating depression. Joint associations were explored by combining SB and each PA into four levels.

Results indicated that SB and OPA were associated with an increased risk of depression. LTPA was

associated with decreased risk only in men, whereas TRPA showed no significant association. In brief, this study suggests that OPA(+), LTPA(–), and TRPA(–) have a synergistic effect on increasing depression risk when concurrently exposed to SB. Encouraging LTPA, minimizing rigorous OPA, and reducing SB may contribute to reducing depressive symptoms.

I would like to than authors for this interesting study. I believe this is an interesting study and it has a merit for this journal. Here is my feedback: This study needs a strong introduction, discussion and conclusion sections with up to date literature review (2022 and above). In addition, reliability and validity of tests and measurements should be reported. In brief, this study has a good design and sample size. However, some parts of this research need improvement. I look forward to seeing edited version of this manuscript. Best regards.

6. PLOS authors have the option to publish the peer review history of their article (what does this mean?). If published, this will include your full peer review and any attached files.

Reviewer #1: No

Reviewer #2: **Yes: **Ferman Konukman

---

## [Editor Report · Decision Letter 1]

30 Sep 2024

Joint association of sedentary behavior and physical activity domains with depression in Korean adults: Cross-sectional study combining four biennial surveys(2016–2022)

PONE-D-24-21184R1

Dear Dr. Park,

We’re pleased to inform you that your manuscript has been judged scientifically suitable for publication and will be formally accepted for publication once it meets all outstanding technical requirements.

Kind regards,

Rashid Menhas, PhD

Academic Editor

PLOS ONE

Additional Editor Comments (optional):

NA
---

## [Editor Report · Acceptance letter]

15 Oct 2024

PONE-D-24-21184R1 

PLOS ONE

Dear Dr. Park, 

I'm pleased to inform you that your manuscript has been deemed suitable for publication in PLOS ONE. Congratulations! Your manuscript is now being handed over to our production team.

Kind regards, 

on behalf of

Dr. Rashid Menhas 

Academic Editor

PLOS ONE